# Paraphrase Generation with Latent Bag of Words

**Yao Fu**
Department of Computer Science
Columbia University
yao.fu@columbia.edu

**Yansong Feng**
Institute of Computer Science and Technology
Peking University
fengyansong@pku.edu.cn

**John P. Cunningham**
Department of Statistics
Columbia University
jpc2181@columbia.edu

## Abstract

Paraphrase generation is a longstanding important problem in natural language processing. In addition, recent progress in deep generative models has shown promising results on discrete latent variables for text generation. Inspired by variational autoencoders with discrete latent structures, in this work, we propose a latent bag of words (BOW) model for paraphrase generation. We ground the semantics of a discrete latent variable by the BOW from the target sentences. We use this latent variable to build a fully differentiable content planning and surface realization model. Specifically, we use source words to predict their neighbors and model the target BOW with a mixture of softmax. We use Gumbel top-k reparameterization to perform differentiable subset sampling from the predicted BOW distribution. We retrieve the sampled word embeddings and use them to augment the decoder and guide its generation search space. Our latent BOW model not only enhances the decoder, but also exhibits clear interpretability. We show the model interpretability with regard to *(i)* unsupervised learning of word neighbors *(ii)* the step-by-step generation procedure. Extensive experiments demonstrate the transparent and effective generation process of this model.[1]

## 1 Introduction

The generation of paraphrases is a longstanding problem for learning natural language [33]. Paraphrases are defined as sentences conveying the same meaning but with different surface realization. For example, in a question answering website, people may ask duplicated questions like *How do I improve my English* v.s. *What is the best way to learn English*. Paraphrase generation is important, not only because paraphrases demonstrate the diverse nature of human language, but also because the generation system can be the key component to other important language understanding tasks, such as question answering[5, 11], machine translation [7], and semantic parsing [43].

Traditional models are generally rule based, which find lexical substitutions from WordNet [34] style resources, then substitute the content words accordingly [3, 36, 21]. Recent neural models primary rely on the sequence-to-sequence (seq2seq) learning framework [44, 40], achieving inspiring performance gains over the traditional methods. Despite its effectiveness, there is no strong interpretability of seq2seq learning. The sentence embedding encoded by the encoder is not directly associated with any linguistic aspects of that sentence[2]. On the other hand, although interpretable, many traditional

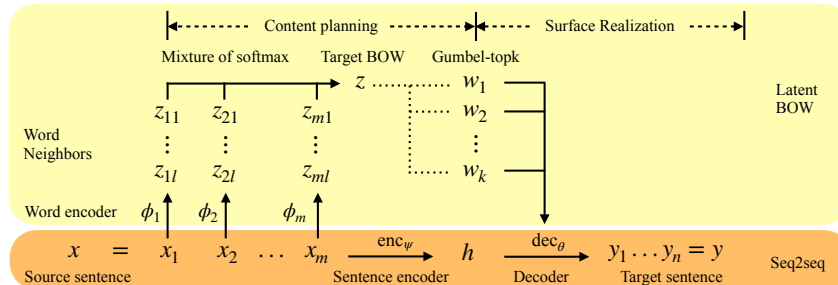

Figure 1: Our model equip the seq2seq model(lower part) with latent bag of words(upper part).

methods suffer from suboptimal performance [40]. In this work we introduce a model with optimal performance that maintains and benefits from semantic interpretability.

To improve model interpretability, researchers typically follow two paths. First, from a probabilistic perspective, one might encode the source sentence into a latent code with certain structures [22] (e.g. a Gaussian variable for the MNIST[24] dataset). From a traditional natural language generation(NLG) perspective, one might explicitly separate *content planning* and *surface realization* [35]. The traditional word substitution models for paraphrase generation are an example of planning and realization: first, word neighbors are retrieved from WordNet (the planning stage), and then words are substituted and re-organized to form a paraphrase (the realization stage). Neighbors of a given word refer to words that are semantically close to the given word (e.g. *improve → learn*). Here the interpretability comes from a linguistic perspective, since the model performs generation step by step: it first proposes the content, then generates according to the proposal. Although effective across many applications, both approaches have their own drawbacks. The probabilistic approach lacks explicit connection between the code and the semantic meaning, whereas for the traditional NLG approach, separation of planning and realization is (across most models) nondifferentiable [6, 35], which then sacrifices the end-to-end learning capabilities of network models, a step that has proven critical in a vast number of deep learning settings.

In an effort to bridge these two approaches, we propose a hierarchical latent bag of words model for planning and realization. Our model uses words of the source sentence to predict their neighbors in the bag of words from target sentences [3]. From the predicted word neighbors, we sample a subset of words as our content plan, and organize these words into a full sentence. We use Gumbel top-k reparameterization[20, 32] for differentiable subset sampling[49], making the planning and realization fully end-to-end. Our model then exhibits interpretability of from both of the two perspectives: from the probabilistic perspective, since we optimize a discrete latent variable towards the bag of words of the target sentence, the meaning of this variable is grounded with explicit lexical semantics; from the traditional NLG perspective, our model follows the planning and realization steps, yet fully differentiable. Our contributions are:

- We endow a hierarchical discrete latent variable with explicit lexical semantics. Specifically, we use the bag of words from the target sentences to ground the latent variable.

- We use this latent variable model to build a differentiable step by step content planning and surface realization pipeline for paraphrase generation.

- We demonstrate the effectiveness of our model with extensive experiments and show its interpretability with respect to clear generation steps and the unsupervised learning of word neighbors.

## 2 Model

Our goal is to extend the seq2seq model (figure 1 lower part) with differentiable content planning and surface realization (figure 1 upper part). We begin with a discussion about the seq2seq base model.

## 2.1 The Sequence to Sequence Base Model.

The classical seq2seq model encodes the source sequence $x = x_1, x_2, ..., x_m$ into a code $h$, and decodes it to the target sequence $y$ (Figure 1 lower part), where $m$ and $n$ are the length of the source and the target, respectively [44, 2]. The encoder $\text{enc}_\psi$ and the decoder $\text{dec}_\theta$ are both deep networks. In our work, they are implemented as LSTMs[18]. The loss function is the negative log likelihood.

$$
\begin{aligned}
h &= \text{enc}_\psi(x) \\
p(y|x) &= \text{dec}_\theta(h) \\
\mathcal{L}_{\text{S2S}} &= \mathbb{E}_{(x^\star, y^\star) \sim \mathbb{P}^\star}[-\log p_\theta(y^\star|x^\star)],
\end{aligned}
\tag{1}
$$

where $\mathbb{P}^\star$ is the true data distribution. The model is trained with gradient based optimizer, and we use Adam [23] in this work. In this setting, the code $h$ does not have direct interpretability. To add interpretability, in our model, we ground the meaning of a latent structure with lexical semantics.

## 2.2 Bag of Words for Content Planning.

Now we consider formulating a plan as a bag of words before the surface realization process. Formally, let $\mathcal{V}$ be the vocabulary of size $V$; then a bag of words (BOW) $z$ of size $k$ is a *random set* formulated as a $k$-hot vector in $\mathbb{R}^V$. We assume $z$ is sampled from a base categorical distribution $p(\tilde{z}|x)$ by $k$ times without replacement. Directly modeling the distribution of $z$ is hard due to the combinatorial complexity, so instead we model its base categorical distribution $p(\tilde{z}|x)$. In paraphrase generation datasets, one source sentence may correspond to multiple target sentences. Our key modeling assumption is that the BOW from target sentences (target BOW) should be similar to the neighbors of the words in the source sentence. As such, we define the base categorial variable $\tilde{z}$ as the mixture of all neighbors of all source words. Namely, first, for each source word $x_i$, we model its neighbor word with a one-hot $z_{ij} \in \mathbb{R}^V$ :

$$
p(z_{ij}|x_i) = \text{Categorical}(\phi_{ij}(x_i))
\tag{2}
$$

The support of $z_{ij}$ is also the word vocabulary $\mathcal{V}$ and $\phi_{ij}$ is parameterized by a neural network. In practice, we use a softmax layer on top of the hidden states of each timesteps from the encoder LSTM. We assume a fixed total number of neighbors $l$ for each $x_i$ ($j = 1, ..., l$). We then mix the probabilities of these neighbors:

$$
\tilde{z} \sim p_\phi(\tilde{z}|x) = \frac{1}{ml} \sum_{i,j} p(z_{ij}|x_i)
\tag{3}
$$

where $ml$ is the maximum number of predicted words. $\tilde{z}$ is a categorical variable mixing all neighbor words. We construct the bag of words $z$ by sampling from $p_\phi(\tilde{z}|x)$ by $k$ times without replacement. Then we use $z$ as the plan for decoding $y$. The generative process can be written as:

$$
\begin{aligned}
z &\sim p_\phi(\tilde{z}|x) \quad \text{(sample } k \text{ times without replacement)} \\
y &\sim p_\theta(y|x, z) = \text{dec}_\theta(x, z)
\end{aligned}
\tag{4}
$$

For optimization, we maximize the negative log likelihood of $p(y|x, z)$ and $p_{\tilde{z}}(\tilde{z}|x)$:

$$
\begin{aligned}
\mathcal{L}_{\text{S2S}'} &= \mathbb{E}_{(x^\star, y^\star) \sim \mathbb{P}^\star, z \sim p_\phi(\tilde{z}|x)}[-\log p_\theta(y^\star|x^\star, z)] \\
\mathcal{L}_{\text{BOW}} &= \mathbb{E}_{z^* \sim \mathbb{P}^*}[-\log p_\phi(z^*|x)] \\
\mathcal{L}_{\text{tot}} &= \mathcal{L}_{\text{S2S}'} + \mathcal{L}_{\text{BOW}}
\end{aligned}
\tag{5}
$$

where $\mathbb{P}^*$ is the true distribution of the BOW from the target sentences. $z^*$ is a $k$-hot vector representing the target bag of words. One could also view $\mathcal{L}_{\text{BOW}}$ as a regularization of $p_\phi$ using the weak supervision from target bag of words. Another choice is to view $z$ as completely latent and infer them like a canonical latent variable model.[4] We find out using the target BOW regularization significantly improves the performance and interpretability. $\mathcal{L}_{\text{tot}}$ is the total loss to optimize over the parameters $\psi, \theta, \phi$. Note that for a given source word in a particular training instance, the NLL loss does not penalize the predictions not included in the targets of this instance. This property makes the model be able to learn different neighbors from different data points, i.e., the learned neighbors will be at a corpus level, rather than sentence level. We will further demonstrate this property in our experiments.

## 2.3 Differentiable Subset Sampling with Gumbel Top-k Reparameterization.

As is discussed in the previous section, the sampling of $z$ (sample k items from a categorical distribution) is non-differentiable. [5] To back-propagate the gradients through $\phi$ in $\mathcal{L}_{S2S'}$ in equation 5, we choose a reparametrized gradient estimator, which relies on the gumbel-softmax trick. Specifically, we perform differentiable subset sampling with the gumbel-topk reparametrization [25]. Let the probability of $\tilde{z}$ to be $p(\tilde{z} = i|x) = \pi_i, i \in \{1, 2, ..., \mathcal{V}\}$, we obtain the perturbed weights and probabilities by:

$$
\begin{aligned}
a_i &= \log \pi_i + g_i \\
g_i &\sim \text{Gumbel}(0, 1)
\end{aligned}
\tag{6}
$$

Retrieving the $k$ largest weights $\text{topk}(a_1, ..., a_{\mathcal{V}})$ will give us $k$ sample *without replacement*. This process is shown in dashed lines in figure 1. We retrieve the $k$ sampled word embeddings $w_1, ..., w_k$ and re-weight them with their probability $\pi_i$. Then we used the average of the weighted word embeddings as the decoder LSTM's initial state to perform surface realization.

Intuitively, in addition to the sentence code $h$, the decoder also takes the weighted sample word embeddings and performs attention[2] to them; thus differentiability is achieved. This generated plan will restrict the decoding space towards the bag of words of the target sentences. More detailed information about the network architecture and the parameters are in the appendix. In section 4, we use extensive experiments to demonstrate the effectiveness of our model.

## 3  Related Work

**Paraphrase Generation.** Paraphrases capture the essence of language diversity [39] and often play important roles in many challenging natural language understanding tasks like question answering [5, 11], semantic parsing [43] and machine translation [7]. Traditional methods generally employ rule base content planning and surface realization procedures [3, 36, 21]. These methods often rely on WordNet [34] style word neighbors for selecting substitutions. Our model can unsupervised learn the word neighbors and predict them on the fly. Recent end-to-end models for paraphrase generation include the attentive seq2seq model[43], the Residual LSTM model [40], the Gaussian VAE model [15], the copy and constrained decoding model [6], and the reinforcement learning approach [26]. Our model has connections to the copy and constrained decoding model by Cao et al. [6]. They use an IBM alignment [9] model to restrict the decoder's search space, which is not differentiable. We use the latent BOW model to guide the decoder and use the gumbel topk to make the whole generation differentiable. Compared with previous models, our model learns word neighbors in an unsupervised way and exhibits a differentiable planning and realization process.

**Latent Variable Models for Text.** Deep latent variable models have been an important recent trend [22, 12] in text modeling. One common path is for researchers to start from a standard VAE with a Gaussian prior [4], which may perhaps encouter issues due to posterior collapse [10, 16]. Multiple approaches have been proposed to control the tradeoff between the inference network and the generative network [52, 50]. In particular, the $\beta-$VAE [17] use a balance parameter $\beta$ to balance the two models in an intuitive way. This approach will form one of our baselines.

Many discrete aspects of the text may not be captured by a continuous latent variable. To better fit the discrete nature of sentences, with the help of the Gumbel-softmax trick [32, 20], recent works try to add discrete structures to the latent variable [53, 48, 8]. Our work directly maps the meaning of a discrete latent variable to the bag of words from the target sentences. To achieve this, we utilize the recent differentiable subset sampling [49] with the Gumbel top-k [25] reparameterization. It is also noticed that the multimodal nature of of text can pose challenges for the modeling process [53]. Previous works show that mixture models may come into aid [1, 51]. In our work, we show the effectiveness of the mixture of softmax for the multimodal bag of words distribution.

**Content Planning and Surface Realization.** The generation process of natural language can be decomposed into two steps: content planning and surface realization (also called sentence generation) [35]. The seq2seq model [44] implicitly performs the two steps by encoding the source sentence into an embedding and generating the target sentence with the decoder LSTM. A downside is that this intermediate embedding makes it hard to explicitly control or interpret the generation process

[2, 35]. Previous works have shown that explicit planning before generation can improve the overall performance. Puduppully et al. [41], Sha et al. [42], Gehrmann et al. [13], Liu et al. [30] embed the planning process into the network architecture. Moryossef et al. [35] use a rule based model for planning and a neural model for realization. Wiseman et al. [48] use a latent variable to model the sentence template. Wang et al. [46] use a latent topic to model the topic BOW, while Ma et al. [31] use BOW as regularization. Conceptually, our model is similar to Moryossef et al. [35] as we both perform generation step by step. Our model is also related to Ma et al. [31]. While they use BOW for regularization, we map the meaning of the latent variable to the target BOW, and use the latent variable to guide the generation.

## 4    Experiments

**Datasets and Metrics.** Following the settings in previous works [26, 15], we use the `Quora`[6] dataset and the `MSCOCO`[28] dataset for our experiments. The `MSCOCO` dataset was originally developed for image captioning. Each image is associated with 5 different captions. These captions are generally close to each other since they all describe the same image. Although there is no guarantee that the captions must be paraphrases as they may describe different objects in the same image, the overall quality of this dataset is favorable. In our experiments, we use 1 of the 5 captions as the source and all content words[7] from the rest 4 sentences as our BOW objective. We randomly choose one of the rest 4 captions as the seq2seq target. The `Quora` dataset is originally developed for duplicated question detection. Duplicated questions are labeled by human annotators and guaranteed to be paraphrases. In this dataset we only have two sentences for each paraphrase set, so we randomly choose one as the source, the other as the target. After processing, for the `Quora` dataset, there are 50K training instances and 20K testing instances, and the vocabulary size is 8K. For the `MSCOCO` dataset, there are 94K training instances and 23K testing instances, and the vocabulary size is 11K. We set the maximum sentence length for the two datasets to be 16. More details about datasets and pre-processing are shown in the appendix.

Although the evaluation of text generation can be challenging [37, 29, 47], previous works show that matching based metrics like BLEU [38] or ROUGE [27] are suitable for this task as they correlate with human judgment well [26]. We report all lower ngram metrics (1-4 grams in BLEU, 1-2 gram in ROUGE) because these have been shown preferable for short sentences [26, 29].

**Baseline Models.** We use the seq2seq LSTM with residual connections [40] and attention mechanism [2] as our baseline (Residual Seq2seq-Attn). We also use the $\beta-$VAE as a baseline generative model and control the $\beta$ parameter to balance the reconstruction and the recognition networks. Since the VAE models do not utilize the attention mechanism, we also include a vanilla sequence to sequence baseline without attention (Seq2seq). It should be noted that although we do not include other SOTA models like the Transformer [45], the Seq2seq-Attn model is trained with 500 state size and 2 stacked LSTM layers, strong enough and hard to beat. We also use a hard version of our BOW model (BOW-hard) as a lower bound, which optimizes the encoder and the decoder separately, and pass no gradient back from the decoder to the encoder. We compare two versions of our latent BOW model: the topk version (LBOW-Topk), which directly chooses the most k probable words from the encoder, and the gumbel version (LBOW-Gumbel), which samples from the BOW distribution with gumbel reparameterization, thus injecting randomness into the model. Additionally, we also consider a cheating model that is able see the BOW of the actual target sentences during generation (Cheating BOW). This model can be considered as an upper bound of our models. The evaluation of the LBOW models are performed on the held-out test set so they cannot see the target BOW. All above models are approximately the same size, and the comparison is fair. In addition, we compare our results with Li et al. [26]. Their model is SOTA on the `Quora` dataset. The numbers of their model are not directly comparable to ours since they use twice larger data containing negative samples for inverse reinforcement learning.[8] Experiments are repeated three times with different random seeds. The average performance is reported. More configuration details are listed in the appendix.

Table 1: Results on the `Quora` and `MSCOCO` dataset. B for BLEU and R for ROUGE.

Quora

| Model | B-1 | B-2 | B-3 | B-4 | R-1 | R-2 | R-L |
|---|---|---|---|---|---|---|---|
| Seq2seq[40] | 54.62 | 40.41 | 31.25 | 24.97 | 57.27 | 33.04 | 54.62 |
| Residual Seq2seq-Attn [40] | 54.59 | 40.49 | 31.25 | 24.89 | 57.10 | 32.86 | 54.61 |
| $\beta$-VAE, $\beta = 10^{-3}$[17] | 43.02 | 28.60 | 20.98 | 16.29 | 41.81 | 21.17 | 40.09 |
| $\beta$-VAE, $\beta = 10^{-4}$[17] | 47.86 | 33.21 | 24.96 | 19.73 | 47.62 | 25.49 | 45.46 |
| BOW-Hard (lower bound) | 33.40 | 21.18 | 14.43 | 10.36 | 36.08 | 16.23 | 33.77 |
| LBOW-Topk (ours) | **55.79** | **42.03** | **32.71** | **26.17** | **58.79** | **34.57** | **56.43** |
| LBOW-Gumbel (ours) | 55.75 | 41.96 | 32.66 | 26.14 | 58.60 | 34.47 | 56.23 |
| RbM-SL[26] | - | 43.54 | - | - | 64.39 | 38.11 | - |
| RbM-IRL[26] | - | 43.09 | - | - | 64.02 | 37.72 | - |
| Cheating BOW (upper bound) | 72.96 | 61.78 | 54.40 | 49.47 | 72.15 | 52.61 | 68.53 |

MSCOCO

| Model | B-1 | B-2 | B-3 | B-4 | R-1 | R-2 | R-L |
|---|---|---|---|---|---|---|---|
| Seq2seq[40] | 69.61 | 47.14 | 31.64 | 21.65 | 40.11 | 14.31 | 36.28 |
| Residual Seq2seq-Attn [40] | 71.24 | 49.65 | 34.04 | 23.66 | 41.07 | 15.26 | 37.35 |
| $\beta$-VAE, $\beta = 10^{-3}$[17] | 68.81 | 45.82 | 30.56 | 20.99 | 39.63 | 13.86 | 35.81 |
| $\beta$-VAE, $\beta = 10^{-4}$[17] | 70.04 | 47.59 | 32.29 | 22.54 | 40.72 | 14.75 | 36.75 |
| BOW-Hard (lower bound) | 48.14 | 28.35 | 16.25 | 9.28 | 31.66 | 8.30 | 27.37 |
| LBOW-Topk (ours) | **72.60** | **51.14** | **35.66** | **25.27** | 42.08 | **16.13** | **38.16** |
| LBOW-Gumbel (ours) | 72.37 | 50.81 | 35.32 | 24.98 | **42.12** | 16.05 | 38.13 |
| Cheating BOW (upper bound) | 80.87 | 75.09 | 62.24 | 52.64 | 49.95 | 23.94 | 43.77 |

## 4.1 Experiment Results

Table 1 show the overall performance of all models. Our models perform the best compared with the baselines. The Gumbel version performs slightly worse than the topk version, but they are generally on par. The margins over the Seq2seq-Attn are not that large (approximately 1+ BLEUs). This is because the capacity of all models are large enough to fit the datasets fairly well. The BOW-Hard model does not perform as well, indicating that the differentiable subset sampling is important for training our discrete latent model. Although not directly comparable, the numbers of RbM models are higher than ours since they are SOTA models on `Quora`. But they are still not as high as the Cheating BOW's, which is consistent with our analysis. The cheating BOW outperforms all other models by a large margin with the leaked BOW information in the target sentences. This shows that the Cheating BOW is indeed a meaningful upper bound and the accuracy of the predicted BOW is essential for an effective decoding process. Additionally, we notice that $\beta-$VAEs are not as good as the vanilla Seq2seq models. The conjecture is that it is difficult to find a good balance between the latent code and the generative model. In comparison, our model directly grounds the meaning of the latent variable to be the bag of words from target sentences. In the next section, we show this approach further induces the unsupervised learning of word neighbors and the interpretable generation stages.

## 5 Result Analysis

### 5.1 Model Interpretability

Figure 2 shows the planning and realization stages of our model. Given a source sentence, it first generates the word neighbors, samples from the generated BOW (planning), and generates the sentence (realization). In addition to the *statistical* interpretability, our model shows clear *linguistical* interpretability. Compared to the vanilla seq2seq model, the interpretability comes from: (1). Unsupervised learning of word neighbors (2). The step-by-step generation process.

**Unsupervised Learning of Word Neighbors.** As highlighted in Figure 2, we notice that the model discovers multiple types of lexical semantics among word neighbors, including: (1). word morphology, e.g., speak - speaking - spoken (2). synonym, e.g., big - large, racket - racquet. (3). entailment,

**Quora**

| Input | why do | people | ask | questions | on | quora | instead of | googling | it |
|---|---|---|---|---|---|---|---|---|---|
| Neighbor | | | *post* | *quora* | | *quora* | | *google* | |
| | | | *answer* | *questions* | | *questions* | | *search* | |
| BOW sample | *ask, quora, people, questions, google, googling, easily, googled, search, answer* | | | | | | | | |
| Output | why do people ask questions on quora that can be easily found on a google search ? | | | | | | | | |

| Input | how do | i | talk | english | fluently | ? | | | |
|---|---|---|---|---|---|---|---|---|---|
| Neighbor | | | *speak* | *english* | *fluently* | | | | |
| | | | *better* | *improve* | *confidence* | | | | |
| BOW sample | *english, speak, improve, fluently, talk, spoken, better, best, confidence* | | | | | | | | |
| Output | how can i improve my english speaking ? | | | | | | | | |

**MSCOCO**

| Input | A | tennis | player | is | walking | while | holding | his | racket |
|---|---|---|---|---|---|---|---|---|---|
| Neighbor | | *court* | *holding* | | *walks* | | *carrying* | | *court* |
| | | *racket* | *man* | | *across* | | *holds* | | *racquet* |
| BOW sample | *holding, man, tennis, walking, racket, court, player, racquet, male, woman, walks* | | | | | | | | |
| Output | A man holding a tennis racquet on a tennis court | | | | | | | | |

| Input | A | big | airplane | flying | in | the | blue | sky | |
|---|---|---|---|---|---|---|---|---|---|
| Neighbor | | *large* | *airplane* | *sky* | | | *blue* | *clear* | |
| | | *large* | *jet* | *airplane* | | | *clear* | *flying* | |
| BOW sample | *blue, airplane, flying, large, plane, sky, clear, air, flies, jet* | | | | | | | | |
| Output | A large jetliner flying through a blue sky | | | | | | | | |

| word morphology | synonym | entailment | metonymy |
|---|---|---|---|

Figure 2: Sentence generation samples. Our model exhibits clear interpretability with three generation steps: (1) generate the neighbors of the source words (2) sample from the neighbor BOW (3) generate from the BOW sample. Different types of learned lexical semantics are highlighted.

e.g., improve - english (4). metonymy[9], e.g., search - googling. The identical mapping is also learned (e.g., blue - blue) since all words are neighbors to themselves.

The model can learn this is because, although without explicit alignment, words from the target sentences are semantically close to the source words. The mixture model drops the order information of the source words and effectively match the predicted word set to the BOW from the target sentences. The most prominent word-neighbor relation will be back-propagated to words in the source sentences during optimization. Consequently, the model discovers the word similarity structures.

**The Generation Steps.** A generation plan is formulated by the sampling procedure from the BOW prediction. Consequently, an accurate prediction of the BOW is essential for guiding the decoder, as is demonstrated by the Cheating BOW model in the previous section. The decoder then performs surface realization based on the plan. During realization, the source of word choice comes from (1). the plan (2). the decoder's language model. As we can see from the second example in Figure 2, the planned words include *english, speak, improve*, and the decoder generates other necessary words like *how, i, my* from its language model to connect the plan words, forming the output: *how can i improve my english speaking?* In the next section, we quantitatively analyze the performance of the BOW prediction and investigate how it is utilized by the decoder.

## 5.2 The Implications of the Latent BOW Prediction

**Distributional Coverage.** We first verify our model effectiveness for the multimodal BOW distribution. Figure 3(left) shows the number of the learned modes during the training process, compared with the number of target modes (number of words in the target sentences). For a single categorical variable in our model, if the largest probability in the softmax is greater than 0.5, we define it as a discovered mode. The figure shows an increasing trend of the mode discovery. In the MSCOCO dataset, after convergence, the number of discovered modes is less than the target modes, while in the Quora dataset, the model learns more modes than the target. This difference comes from the two different aspects of these datasets. First, the MSCOCO dataset has more target sentences (4 sentences) than the Quora dataset (1 sentence), which is intuitively harder to cover. Second, the MSCOCO dataset has a noisier nature because the sentences are not guaranteed to be paraphrases. The words in the

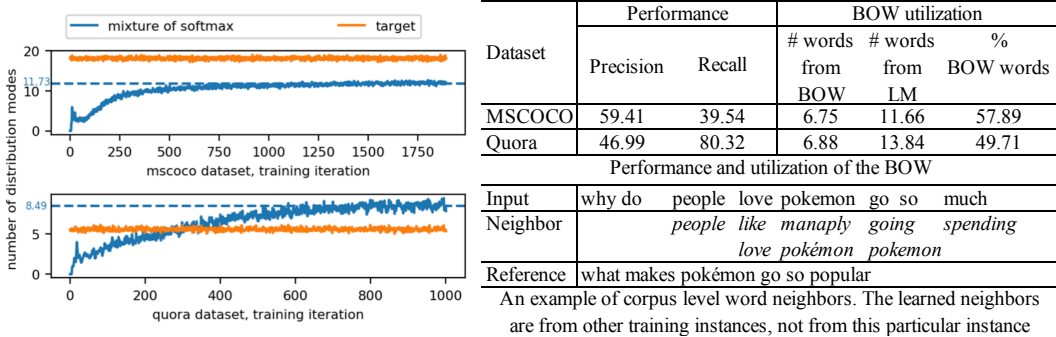

| Dataset | Performance | | BOW utilization | | |
|---|---|---|---|---|---|
| | Precision | Recall | # words from BOW | # words from LM | % BOW words |
| MSCOCO | 59.41 | 39.54 | 6.75 | 11.66 | 57.89 |
| Quora | 46.99 | 80.32 | 6.88 | 13.84 | 49.71 |
| Performance and utilization of the BOW | | | | | |

| Input | why do | people love pokemon go so | much |
|---|---|---|---|
| Neighbor | | *people like manaply going spending* | |
| | | *love pokémon pokemon* | |
| Reference | what makes pokémon go so popular | | |

An example of corpus level word neighbors. The learned neighbors are from other training instances, not from this particular instance

Figure 3: The effectiveness of learned BOW. Left: the learned modes v.s. the average modes in the bag of words from the target sentences. Our model effectively estimates the multimodal BOW distribution. Right upper: BOW prediction performance and utilization. The model heavily uses the predicted BOW, indication the BOW prediction accuracy is essential to a good generation quality. Right lower: an example of corpus level word neighbors. The model learns neighbor words from different instances across the whole training set.

Exploit the BOW information with different components. Adding more sophisticated techniques to the BOW yields consistent improvements

| Quora | B1 | B2 | R1 | R2 |
|---|---|---|---|---|
| seq2seq | 54.6 | 40.41 | 57.27 | 33.04 |
| LBOW | 55.8 | 42.03 | 58.79 | 34.57 |
| LBOW + BOW emb | 56.16 | 42.14 | 58.66 | 34.36 |
| LBOW + Copy | 56.53 | 42.67 | 59.85 | 35.30 |

| Input | A man on | | a | motorcycle with a | bird on the handle |
|---|---|---|---|---|---|
| BOW sample 1 | man | motorcycle *sitting* | | | |
| Output 1 | A man is | | *sitting* on | a | motorcycle |
| BOW sample 2 | man | motorcycle *riding road* | | | |
| Output 2 | A man *riding* | | a | motorcycle on | a | dirt *road* |

| Input | A man wearing | | a | red tie | holding | it | to show people |
|---|---|---|---|---|---|---|---|
| BOW sample 1 | man | suit | tie | | | | |
| Output 1 | A man wearing | | a | suit | and tie | | |
| BOW sample 2 | man | suit | tie | *holding* | *picture* | | |
| Output 2 | A man wearing | *a* | suit | and tie is | *holding* a picture | | |

Figure 4: Left: adding more modeling techniques will consistently improve the model performance. Right: interpolating the latent space. The control of the output sentence can be achieved by explicitly choose or modify the sampled bag of words (in italic).

target sentences might not be as strongly correlated with the source. For the Quora dataset, since the NLL loss does not penalize modes not in the label, the model can discover the neighbor of a word from different context in multiple training instances. In figure 3 right lower, word neighbors like *pokemon-manaply, much-spending* are not in the target sentence, they are generalized from other instances in the training set. In fact, this property of the NLL loss allows the model to learn the corpus level word similarity (instead of the sentence level), and results in more predicted word neighbors than the BOW from one particular target sentence.

**BOW Prediction Performance and Utilization.** As shown in Figure 3 (right), the precision and recall of the BOW prediction is not very high (39+ recall for MSCOCO, 46+ precision for Quora). The support of the precision/ recall correspond the to number of predicted/ target modes respectively in the left figure. We notice that the decoder heavily utilizes the predicted words since more than 50% of the decoder's word choices come from the BOW. If the encoder can be accurate about the prediction, the decoder's search space would be more effectively restricted to the target space. This is why leaking the BOW information from the target sentences results in the best BLEU and ROUGE scores in Table 1. However, although not being perfect, the additional information from the encoder still provides meaningful guidance, and improves the decoder's overall performance. Furthermore, our model is orthogonal to other techniques like conditioning the decoder's each input on the average BOW embedding (BOW emb), or the Copy mechanism[14] (copy). When we integrate our model with such techniques that better exploit the BOW information, we see consistent performance improvement (Figure 4 left).

## 5.3 Controlled Generation through the Latent Space

One advantage of latent variable models is that they allow us to control the final output from the latent code. Figure 4 shows this property of our model. While the interpolation in previous Gaussian

VAEs [24, 4] can only be interpreted as the arithmetic of latent vectors from a geometry perspective, our discrete version of interpolation can be interpreted from a lexical semantics perspective: adding, deleting, or changing certain words. In the first example, the word *sitting* is changed to be *riding*, and one additional *road* is added. This results the final sentence changed from *man ... sitting ...* to *man riding ... on ... road*. The second example is another addition example where *holding, picture* are added to the sentence. Although not quite stable in our experiments[10], this property may induce further application potential with respect to lexical-controllable text generation.

## 6 Conclusion

The latent BOW model serves as a bridge between the latent variable models and the planning-and-realization models. The interpertability comes from the clear generation stages, while the performance improvement comes from the guidance by the sampled bag of words plan. Although effective, we find out that the decoder heavily relies on the BOW prediction, yet the prediction is not as accurate. On the other hand, when there exists information leakage of BOW from the target sentences, the decoder can achieve significantly higher performance. This indicates a future direction is to improve the BOW prediction to better restrict the decoder's search space. Overall, the step by step generation process serves an move towards more interpretable generative models, and it opens new possibilities of controllable realization through directly injecting lexical information into the middle stages of surface realization.

### Acknowledgments

We thank the reviewers for their detailed feedbacks and suggestions. We thank Luhuan Wu and Yang Liu for the meaningful discussions. This research is supported by China Scholarship Council, Sloan Fellowship, McKnight Fellowship, NIH, and NSF.

## Footnotes

[1] Our code can be found at https://github.com/FranxYao/dgm_latent_bow

[2] The linguistic aspects we refer include but not limited to: words, phrases, syntax, and semantics.

[3]In practice, we gather the words from target sentences into a set. This set is our target BOW.

[4]In such case, one could consider variational inference (VI) with $q(z|x)$ and regularize the variational posterior. Similar with our relaxation of $p(z|x)$ using $p(\tilde{z}|x)$, there should also be certain relaxations over the variational family to make the inference tractable. We leave this to future work.

[5]One choice could be the score function estimator, but empirically it suffers from high variance.

[6]https://www.kaggle.com/aymenmouelhi/quora-duplicate-questions

[7] We view nouns, verbs, adverbs, and adjectives as content words. We view pronouns, prepositions, conjunctions and punctuation as non-content words.

[8]They do not release their code so their detailed data processing should also be different with ours, making the results not directly comparable.

[9]Informally, if A is the metonymy of B, then A is a stand-in for B, e.g., the White House - the US government; Google - search engines; Facebook - social media.

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
