[Supplementary Material]

# Paraphrase Generation with Latent Bag of Words Appendix

## 1 Experiments setup

### 1.1 Data Preprocessing

**The `Quora` Dataset.** This dataset is originally used for duplicated question detection [1]. The original dataset labels question pairs as duplicated or not. We extract all pairs labeled as duplicated. We use 50K of the extracted pairs as the training set, 20K as the test set, and abandon the rest. We truncate the maximum sentence length to be 16. We set word with occurrence less than 5 to be "UNK". Only content words (nouns, verbs, adverbs, and adjectives) are used in bag of words prediction. Non-content words (pronouns, prepositions, conjunctions, and punctuation) are not used in the latent BOW model. The maximum size of the target BOW set is 11. The sample size is also 11. Larger sample size will give the decoder more flexibility about word choice, while smaller sample size will give the decoder more certainty. The vocabulary size is 8K. Compared with the `MSCOCO` dataset, the upside of this dataset is that sentences from this dataset are labeled to be paraphrases by human annotators. The downside of this dataset is that we only have one target sentence, while the `MSCOCO` dataset has multiple target sentences. Despite the downside, this dataset is informative enough for the model to learn meaningful paraphrases.

**The `MSCOCO` Dataset.** This dataset is originally used for image captioning. The sentences are written by human annotators to describe the objects in the image. There are five annotations for each image. Generally, these annotations are paraphrases to each other because they all describe the same image. However, they are not guaranteed to be paraphrases because different annotations may describe different objects in the image, e.g. if there is a table and a chair in the image, one caption may be about the table, another may be about the chair. In training, we loop over the five sentence, using the $i$th as the source sentence, the $i+1$th as the target sentence, and all words in the 5 sentences except the source words as the target BOW. The maximum sentence length is 16. The maximum size of the target BOW is 25, and we sample 10 words from the predicted BOW. In evaluation, we use one of the five sentences as the input, and the rest four sentences as the references for computing BLEU and ROUGE. There are 94K training instances and 23K testing instances. The vocabulary size is 11K. Although sentences are not guaranteed to be paraphrases, they generally are. So this is also a meaningful dataset for the paraphrase generation task.

### 1.2 Model Architecture, Hyper-parameters, and Training

For the encoder, we use a two-layer residual LSTM. The decoder is also a two-layer residual LSTM. We find out that the residual connections significantly speed up the convergence. Each LSTM's state size is 500, and so is the word embedding size. The prediction of word neighbor is similar to a sequence tagging model. For each source word, we use three separate softmax layers on top of the LSTM output to predict their three neighbors. The three softmax may output the same neighbor, if

there is not so many neighbors in the training set. The number of neighbors for a given word in a dataset is an intrinsic property of this dataset, and our model discovers this in an unsupervised way.

Sampling from the predicted neighbor is achieved by the gumbel-topk trick. The temperature parameter of the gumbel-topk is set to be 1. The embeddings of the sampled words are retrieved from the embedding table, and re-weighted by their corresponding probability in the bag of words distribution. The weighted average of the word embeddings are then added to the encoder's last hidden state, and used as the decoder's initial hidden state. During decoding, the decoder not only perform attention to the source words as a normal attentive seq2seq model, but also perform attention to the sampled BOW embeddings. We use greedy decoding in all the models.

We use the Adam optimizer for training. The learning rate is set to be 0.0008 for all models, and this number seems to be the best learning rate from our hyper-parameter search. We set dropout to be 0.6 since the model is large enough for the datasets. We find out the dropout strength and the learning rate are the two most influential hyper-parameters. We set the batch size to be 100. Each model are trained with 10 epochs, and the models with the best performance are chosen. All experiments are repeated three times with different random seeds. The average performance of the experiments are reported. The standard deviation of the numbers are smaller enough than our models' performance gain.

Since the previous works [Li et al., 2018, Prakash et al., 2016, Gupta et al., 2018] do not release their code, we are not sure how exactly they process the data, and we cannot reproduce exactly the same results. Since our preprocessing of the dataset is different than their setting. Our reported numbers are not directly comparable to theirs. However, they are meaningful in our setting and comparable to each other.

## 2 Generation Samples

Figure 1 gives more generated sentences from the `Quora` dataset. Figure 2 gives more sentences from the `MSCOCO` dataset. Not all predicted word neighbors are meaningful and not all generated sentences are successful. But still, the overall generation quality is favorable.

Figure 3 gives more sample about controlling through the latent code. This property is not quite stable in our experiments. Changing a sampled word does not necessarily change the final output sentence since the decoder may ignore the BOW sample. Also the change of the word choice can not influence multiple words (more than three words) at present. In our experiments, we see more successful results about changing one or two words. Despite the instability, this property gives a new possibility about controllable text generation through the intermediate word choice. A future direction is to make this property more stable and controllable.

## Footnotes

[1]https://www.kaggle.com/aymenmouelhi/quora-duplicate-questions

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

| Input | what | is | the | best | path | i | should | take | to | improve | english |
|---|---|---|---|---|---|---|---|---|---|---|---|
| Neighbor | | | | *best* | *way* | | | | | *learn* | *improve* |
| | | | | *good* | *want* | | | | | *improve* | *speaking* |
| BOW sample | *improve, best, english, way, spoken, learn, speaking, communication, pronunciation, good* |
| Output | how can i improve my english language ? |

| Input | what | are | the | popular | series | by | unk | ? |
|---|---|---|---|---|---|---|---|---|
| Neighbor | | | | *popular* | *watch* | | | |
| | | | | *best* | *movies* | | | |
| BOW sample | *series, popular, best, watch, tv 's, movies, kind, famous* |
| Output | what are some of the best movies by unk ? |

| Input | do | you | think | there | is | life | on | other | planets | ? |
|---|---|---|---|---|---|---|---|---|---|---|
| Neighbor | | | *think* | | | *life* | | *believe* | *know* | |
| | | | *think* | | | *life* | | *signs* | *found* | |
| BOW sample | *life, think, planets, found, believe, know, sustain, signs* |
| Output | do you believe in life on other planets ? |

| Input | what | is | the | procedure | to | check | your | pf | balance | ? |
|---|---|---|---|---|---|---|---|---|---|---|
| Neighbor | | | | *procedure* | | *check* | | *check* | *account* | |
| | | | | *open* | | *know* | | *offline* | *report* | |
| BOW sample | *pf, check, balance, procedure, know, offline, report, account, profile, requirements, process* |
| Output | how do i check my pf balance ? |

| Input | what | role | does | money | play | in | your | life | ? |
|---|---|---|---|---|---|---|---|---|---|
| Neighbor | | *role* | | *much* | *play* | | *life* | | |
| | | *role* | | *money* | *played* | | *life* | | |
| BOW sample | *life, play, role, money, played, like, would, importance* |
| Output | what is the role of money in your life ? |

| Input | what | is | the | bad | thing | about | quora | ? |
|---|---|---|---|---|---|---|---|---|
| Neighbor | | | | *bad* | *thing* | | *quora* | |
| | | | | *bad* | *things* | | *quora* | |
| BOW sample | *quora, bad, thing, things,  best, answer, life, good* |
| Output | what is the most thing to quora ? |

Figure 1: More sentence generation samples from the Quora dataset

| Input | Two | people | walking | their | dogs | on | a | beach | | | |
|---|---|---|---|---|---|---|---|---|---|---|---|
| Neighbor | | *men* | *walking* | | *dogs* | | | *sand* | | | |
| | | *couple* | *walk* | | *dog* | | | *beach* | | | |
| BOW sample | *people, beach, walking, water, couple, dogs, carrying, walk, ocean* | | | | | | | | | | |
| Output | Two people walking on the beach with their dogs | | | | | | | | | | |

| Input | A | woman | eating | a | piece | of | pastry | in | a | market | area |
|---|---|---|---|---|---|---|---|---|---|---|---|
| Neighbor | | *woman* | *food* | | *slice* | | *donut* | | | *store* | *street* |
| | | *young* | *bite* | | *eating* | | *doughnut* | | | *shopping* | *outdoor* |
| BOW sample | *eating, woman, bite, doughnut, pastry, donut, taking, holding, piece, slice, food* | | | | | | | | | | |
| Output | A woman eating a piece of food in a market | | | | | | | | | | |

| Input | There | is | a | bus | with | several | people | standing | next | to | it |
|---|---|---|---|---|---|---|---|---|---|---|---|
| Neighbor | | | | *transit* | | *driving* | *people* | *people* | *bus* | | |
| | | | | *parked* | | *street* | *passengers* | *standing* | *parked* | | |
| BOW sample | *bus, people, parked, standing, next, street, waiting, passengers* | | | | | | | | | | |
| Output | A group of people standing next to a bus | | | | | | | | | | |

| Input | A | man | on | a | snow | board | jumps | off | a | snow | hill |
|---|---|---|---|---|---|---|---|---|---|---|---|
| Neighbor | | *man* | | | *snowy* | *snowboard* | *air* | | | *hill* | *slope* |
| | | *person* | | | *slope* | *board* | *jumpping* | | | *snowboard* | *hill* |
| BOW sample | *man, jumping, air, snow, snowboard, board, person, hill, jump, snowboarder* | | | | | | | | | | |
| Output | A man on a snowboard jumping over a hill | | | | | | | | | | |

| Input | A | large | propeller | airplane | flying | through | a | foggy | sky | | |
|---|---|---|---|---|---|---|---|---|---|---|---|
| Neighbor | | *large* | *propeller* | *airplane* | *sky* | | | *fog* | *flying* | | |
| | | *large* | *airplane* | *propeller* | *plane* | | | *cloudy* | *clear* | | |
| BOW sample | *flying, plane, airplane, air, sky, large, propeller, cloudy, flies, blue* | | | | | | | | | | |
| Output | A large airplane flying through a cloudy sky | | | | | | | | | | |

| Input | A | group | of | young | people | playing | video | games | | | |
|---|---|---|---|---|---|---|---|---|---|---|---|
| Neighbor | | *group* | | *people* | *people* | *holding* | *remote* | *video* | | | |
| | | *group* | | *young* | *group* | *standing* | *room* | *game* | | | |
| BOW sample | *game, group, people, video, playing, young, Wii, standing, Nintendo, games* | | | | | | | | | | |
| Output | A group of people playing a game with remote controllers | | | | | | | | | | |

Figure 2: More sentence generation samples from the MSCOCO dataset

| | |
|---|---|
| Input | A motorcycle parked in a parking space next to another motorcycle |
| BOW sample 1 | motorcycle  lot |
| Output 1 | A motorcycle parked in a parking lot next to a building |
| BOW sample 2 | motorcycle  lot          *another* |
| Output 2 | A motorcycle parked next to *another* motorcycle in a parking lot |

| | |
|---|---|
| Input | A baseball player is swinging at a baseball |
| BOW sample 1 | baseball      people |
| Output 1 | A baseball player is swinging at a ball |
| BOW sample 2 | baseball      people      *ready* |
| Output 2 | A baseball player is getting *ready* to catch a ball |

| | |
|---|---|
| Input | A group of young people playing video games |
| BOW sample 1 | playing      group      game      *men* |
| Output 1 | A group of young *men* playing a game with Nintendo Wii controllers |
| BOW sample 2 | playing      group      game      *people* |
| Output 2 | A group of *people* playing a game with Nintendo Wii controllers |

| | |
|---|---|
| Input | A teenaged boy prepares to hit a tennis ball with a forehand shot |
| BOW sample 1 | tennis        *racquet* |
| Output 1 | A young boy holding a tennis *racquet* on a tennis court |
| BOW sample 2 | tennis        holding    *racket*      *ball* |
| Output 2 | A young boy is holding a tennis *racket* and *ball* |

| | |
|---|---|
| Input | A brown bear looking up to the sky while smiling |
| BOW sample 1 | standing      bear        brown        *field* |
| Output 1 | A brown bear standing on top of a lush green *field* |
| BOW sample 2 | brown        bear        *looking* |
| Output 2 | A brown bear standing in the grass *looking* at something |

| | |
|---|---|
| Input | people built a dog house shaped sand castle |
| BOW sample 1 | dog          men          sitting |
| Output 1 | A dog is standing in front of a large building |
| BOW sample 2 | people      men          dog          People |
| Output 2 | A dog is standing in front of a large building |

Figure 3: More sentence samples from the latent code.