[Reviews · NeurIPS 2019]

Reviewer 1



Thus paper presents a model where a latent bag-of-words inform a paraphrase generation model. For each source words, the authors compute a multinomial over "neighbor" vocabulary words; this then yields a bag-of-words by a mixture of softmaxes over these neighbors. In the generative process, a set of words is drawn from this distribution, then their word embeddings are averaged to form input to the decoder. During training, the authors use a continuous relaxation of this with Gumbel top-k sampling (a differentiable way to sample k of these words without replacement). The words are averaged and fed into the LSTM's initial state. Results show decent BLEU and ROUGE scores compared to baselines as well as some nice examples. However, the authors don't compare against any baselines from prior work, instead comparing against their own implementations of basic Seq2seq, Seq2seq + Attn, and VAE models. As a result, it's a bit hard to situate the results with respect to prior efforts. As for the model itself, I like the premise a lot but I am a bit disappointed by its actual implementation. Averaging the word embeddings to get an initial state for the decoder seems like "giving up" on the fact that you actually have a bag-of-words that generation should be conditioned on. It would be much more interesting to at least attend over this collection or ideally use some more complex generative process that respects it more heavily. In light of this, I would also like to see a comparison to a model that simply treats the phi_ij as mixture weights, then computes the input to the decoder by summing word vectors together. As it stands, I'm not sure how much value the top-k sampling layer is giving, since it immediately gets smashed back down into a continuous representation again. I do see the value in principle, but this and other ablations would convince me more strongly of the model's benefit. Based on Figure 2, I'm not really convinced the sample is guiding the generation strongly as opposed to providing a latent topic. The bag-of-words is definitely related to the output, but not very closely. The Figure 4 results are more compelling, but I'd have to see more than these two examples to be truly convinced. Overall, I like the direction the authors are going with this, but I'm not quite convinced by the experimental evaluation and I think the model could be more than it is currently. ======================== Thanks for the author response; I have left the review above unchanged, but I provide some extra comments here. I see now that this is in the supplementary material, but it is still unclear from the main paper. In light of this, I have raised my score to a 7. I think this model is quite nice for this task. The results are still a bit marginal, but stronger given what's shown in Table 4 in the response. Finally, the comparisons in Table 1, even if not totally favorable, at least situate the work with respect to prior efforts. So this has largely addressed my criticisms.

Reviewer 2



This paper proposes an interesting and novel method for paraphrase generation. I enjoyed reading the paper: the description is clear, related work is balanced, and result analysis section is convincing. Minor suggestions: - Figure 1 could be expanded with more details, tied to the equations or sections in Sec 2. - The term CBOW invokes a standard word embedding method as opposed to "cheating BOW". It's fine, but confused me for a bit originally. - If space permits, some more details about the Gumbel implementation would be helpful. == Note after author response == I think my suggestions weren't mentioned in the author response since they were minor points (which is fine), but I trust the authors will improve the draft in the revision.

Reviewer 3



The paper presents a simple fully differentiable discrete latent variable model for content planning and surface realization for sentence-level paraphrase generation. The discrete latent variables are grounded to the BOW from the target sentences bringing semantic interpretability to the latent model. The paper is very well written and the proposed approach is thoroughly evaluated on the sentence-level paragraph generation with both quantitative and qualitative analysis. One of my main concern with the approach is its evaluation on the sentence-level paragraph generation. The need for content planning for sentence-level paragraph generation is rather limited, often we aim to generate the target sentence which is semantically equivalent to the input sentence, there is no need for content selection or content reordering during content planning. It is no surprise that the simple model just as the BOW model is good enough here. The decoder simply takes an average representation of the bag of the words and generates the target sentence. I am afraid that the presented model will not influence or be useful for more complex text generation tasks such as document summarization or data-to-text generation. In these tasks, they often require more sophisticated content selection and content reordering during content planning. What are the state of the art results on the Quora and Mscoco datasets? It is not surprising that the Seq2seq-Attn model seems to be doing well as well. The attention scores also learns a semantic interpretation of the information that is used during decoding. It would have been interesting to see the outputs of the Seq2seq-Attn model along with the LBOW models to understand how is the unsupervised neighbour words learning useful.

[Author Response · NeurIPS 2019]

**Paper 7578: Paraphrase Generation with Latent Bag of Words**

We thank all reviewers for their detailed constructive feedback and suggestions.

**Major concerns/clarifications:**

- **Clarifying a critical error:** First, we have noticed that both Reviewer 1 and Reviewer 2 suggest that the latent BOW is merely taking an average representation of the bag of words as the decoder initial state. We emphasize that this is **not correct**, and we apologize for our paper leading to this misunderstanding. Critically, the decoder also performs attention to the BOW (appendix line 42-43, source code codes/src/latent_bow.py line 207), precisely as requested by Reviewer 1. We will clarify this in the paper.

- **New results/model enhancements further to Reviewer 1's and Reviewer 3's main concern**: We agree that a "more complex generative process" would enhance the paper. Accordingly, to better exploit the BOW information, we now condition the decoder's inputs on the mixed BOW embeddings (with $z_{ij}$), and further integrate the *Copy Mechanism* (Gu et al., 2016; See et al., 2017), directly copying a word from BOW as the output. These mechanisms all yield constant improvements (Table 4). We will also update the results section in the paper and release the code.

We think these two enhancements (and the clarity around our existing use of attention) improve the paper considerably, and we ask the reviewers to reconsider the contributions in light of this clarification and enhancement.

**Additional important concerns:**

- **Comparison with previous works (reviewer 1 and 3):** Thank you; this is an important point, and we have improved the paper considerably on this point. First, although we did not mention this explicitly, our baseline model, seq2seq-attn has basically *identical architecture* as the Residual LSTM (Prakash et al., 16). On the quora dataset, the SOTA model is RbM with inverse reinforcement learning (Li et al., 2018). Since they do not release the code, we list our implementation results and theirs reported on Table 1. Generally we have close numbers. Their model has better scores than ours, which may come from (a) they use twice the size of training set, (b) they directly optimize the BLEU and ROUGE scores. Our advantages are the model transparency and interpretability. On the MSCOCO dataset, the baseline model is Prakash et al (2016), but without released code. We are unsure about many details (train-test split, BLEU ngrams etc.). The experiments in our paper are on MSCOCO17, and Prakash et al (2016) is on MSCOCO14. So we redo our experiments on MSCOCO14 and try to make the settings as comparable as possible, with the results in Table 2. Generally we have comparable numbers. Also we will release all implementations in an effort to establish a fair comparison for future research.

- **More samples (reviewer 1 and 3):** The comparison between LBOW and seq2seq is listed in Table 3. Generally our model has better word choice because of the BOW. More samples from our model are in the appendix.

- **The paraphrase task itself (reviewer 3):** We view paraphrase generation as a reliable benchmark task since it also requires meaningful word choice and ordering, and hence it is our focus in this work. We agree other tasks like data-to-text are challenging and important, so on your recommendation we are now implementing the experiments on the Wikibio dataset(Lebret et. al. 16). Our preliminary results (table 5) have close numbers with SOTA model (Li et. al. 18) and indicate the value of this model in that task as well; we will complete the results for publication.

**Table 1.** Quora results comparison between ours and the SOTA (Li .et .al 18), dispite different implementations, the numbers are close

| Li .et .al (18) | R2 | B2 | Our Implementation | R2 | B2 |
|---|---|---|---|---|---|
| Seq2seq | 31.47 | 36.55 | Seq2seq | 33.04 | 40.41 |
| Residual LSTM | 32.43 | 37.38 | Residual LSTM | 32.86 | 40.49 |
| RbM-SL | 38.11 | 43.54 | LBOW-topK | 34.57 | 42.03 |
| RbM-IRL | 37.72 | 43.09 | LBOW-gumbel | 34.47 | 41.96 |

**Table 2.** MSCOCO 14 results compared with the baseline. The implementation details of the baseline model are unclear. If the bleu reported by Prakash .et.al (16) is bleu3, then we have close numbers

| Prakash .et .al (16) | Bleu | Our implementation | Bleu3 |
|---|---|---|---|
| seq2seq-attn (vanilla) | 33.1 | seq2seq-attn (vanilla) | 33.94 |
| seq2seq-attn (residual) | 37.0 | seq2seq-attn (residual) | 33.96 |
| - | - | LBOW | 35.71 |
| 4 layer lstm, 512 hidden, 0.5 dropout, bleu ngram unspecified | | 4 layer lstm, 512 hidden, 0.5 dropout, bleu 3 | |

**Table 3.** Model ourputs comparison. Our model generally has a better word choice due to the predicted BOW.

| | |
|---|---|
| Input | what are some ways to build your blog audience |
| S2S-Attn | how do i create a blog |
| LBOW | how do i build my blog audience |
| Input | can you name great works of art inspired by atheism |
| S2S-Attn | can you the art of mind |
| LBOW | can you name a great name of atheism |
| Input | is there somewhere i can host my django web app |
| S2S-Attn | can i host my app |
| LBOW | how can i host my web app |
| Input | what are the best ways to build up my credit score |
| S2S-Attn | what are some ways to build up with a credit |
| LBOW | how do i build up credit score |

**Table 4.** Exploit the BOW information with different components. Adding more sophisticated techniques to the BOW yields consistent improvements

| Quora | B1 | B2 | R1 | R2 |
|---|---|---|---|---|
| seq2seq | 54.62 | 40.41 | 57.27 | 33.04 |
| seq2seq-attn | 54.59 | 40.49 | 57.1 | 32.86 |
| LBOW | 55.79 | 42.03 | 58.79 | 34.57 |
| LBOW + BOW emb | 56.16 | 42.14 | 58.66 | 34.36 |
| LBOW + Copy | 56.53 | 42.67 | 59.85 | 35.30 |

**Table 5.** prelimiary results on data to text genetaion. Our method shows improvements and has comparable numbers with SOTA

| Our Implementation | B4 | R2 |
|---|---|---|
| Seq2seq-attn | 40.82 | 52.48 |
| LBOW + Copy | 42.00 | 53.55 |
| Liu et.al.(18) | B4 | R2 |
| Seq2seq-attn | 43.65 | - |
| Structure-aware S2S | 44.89 | - |

[Meta-Review · NeurIPS 2019]

The paper proposes a two-stage model for sentence-level paraphrase generation, trained end-to-end. The first stage is content planning (specifically predicting a 'latent' bag of keywords). The second one is the surface realization stage (forming a sentence relying on the keywords). The model is interesting and novel. The evaluation is sufficiently convincing (the author response, I believe, addressed initial concerns of the reviewer 1). There is a consensus between the reviewers that the work should be accepted. I find the approach creative and think that it may have an impact on other text generation tasks, beyond paraphrasing. I encourage the authors to address the reviewers' comments and incorporate information presented in the author response (e.g., see R2's comment regarding the presentation and details about the baselines). [This meta-review was reviewed and revised by the Program Chairs]